# Blockchain-Enabled Supply Chain platform for Indian Dairy Industry: Safety and Traceability

**DOI:** 10.3390/foods11172716

**Published:** 2022-09-05

**Authors:** Abhirup Khanna, Sapna Jain, Alessandro Burgio, Vadim Bolshev, Vladimir Panchenko

**Affiliations:** 1Department of Systemics, School of Computer Science, University of Petroleum and Energy Studies, Dehradun 248007, India; 2Department of Applied Sciences and Humanities (Chemistry), University of Petroleum and Energy Studies, Bidholi, Energy Acres, Dehradun 248007, India; 3Independent Researcher, 87036 Rende, Italy; 4Federal Scientific Agroengineering Center VIM, Moscow 109428, Russia; 5Department of Theoretical and Applied Mechanics, Russian University of Transport, Moscow 127994, Russia

**Keywords:** dairy products, food safety, traceability, supply chain management, blockchain

## Abstract

Conventional food supply chains are centralized in nature and possess challenges pertaining to a single point of failure, product irregularities, quality compromises, and loss of data. Numerous cases of food fraud, contamination, and adulteration are daily reported from multiple parts of India, suggesting the absolute need for an upgraded decentralized supply chain model. A country such as India, where its biggest strength is its demographic dividend, cannot afford to malnutrition a large population of its children by allowing them to consume contaminated and adulterated dairy products. In view of the gravity of the situation, we propose a blockchain-enabled supply chain platform for the dairy industry. With respect to the supply chain platform, the dairy products of choice include milk, cheese, and butter. Blockchain is one of the fastest growing technologies having widespread acceptance across multiple industry verticals. Blockchain possesses the power to transform traditional supply chains into decentralized, robust, transparent, tamper proof, and sustainable supply chains. The proposed supply chain platform goes beyond the aspect of food traceability and focuses on maintaining the nutritional values of dairy products, identification of adulteration and contamination in dairy products, the increasing economic viability of running a dairy farm, preventing counterfeit dairy products, and enhancing the revenue of the dairy company. The paper collates the mentioned functionalities into four distinct impact dimensions: social, economic, operations, and sustainability. The proposed blockchain-enabled dairy supply chain platform combines the use of smart contracts, quick response code (QR code) technology, and IoT and has the potential to redefine the dairy supply chains on socio-economic, operational, and sustainability parameters.

## 1. Introduction

Food constitutes one of the most important aspects of human life. Since almost a millennium, humans have been fascinated by the concept of food and from mere mortal pleasure, it has evolved to be a source of goodwill and happiness. Humans consume food on the belief that the products they are consuming are manufactured, processed, stored, and transported in ways that are following the quality standards. However, in the past couple of years, thousands of humans across the world have been infected by consuming contaminated and adulterated food products. One such category of food is dairy products, which have seen significant contamination and adulteration practices in the past. The history of milk adulteration can be traced to 1850 when in New York about 8000 children were killed by the Swill milk scandal [1]. Later, it became a serious concern when in China infant milk products were adulterated with melamine [2]. The adulteration is made for economic reasons; however, it affects public health [3]. Dairy products are one of the largest consumable categories of food products and are seen as a primary source of nutrition across multiple age groups. However, adulteration and contamination affect the nutritional value and host presence of perilous substances making milk inconsumable and harmful. The common milk adulterants are water, starch, urea, glucose, detergents, Vanaspati, and preservatives. Contamination can be biological (microbes), chemical (pesticides, antibiotics, metal etc.), or physical (dust). The contamination in milk initiates in the mammary glands of livestock by excretion of xenobiotic substances such as antibiotics and other veterinary drugs followed by exposure to environmental pollutants, pesticides, pathogens, etc. Fodder is also a significant source of contamination as it may increase the spore load in raw milk. A bad hygiene practice is primarily responsible for microbial contamination in milk, cheese, butter, and other milk products. A knowledge of udder health status, antibiotics administrated, and common pathogens in the dairy farm is a must to ensure milk safety and dairy product quality. The other microbial contamination is possible during the long storage period, inefficient cooling storage practices, use of non-sterilized storage tanks, etc. Poor pasteurization paves the path for the survival of microbial pathogens and may cause several food-borne diseases.

Adulteration starts from the mere addition of water into milk. The addition of water decreases the nutritional value of milk and can cause several diseases. The food chain of a dairy product must be transparent to its customers and other participants to maintain the prescribed quality and nutritional values of a product. The lack of transparency, traceability, and provenance in the dairy supply chain has led to large-scale food frauds impacting the lives of millions. The dairy supply chain is one of the most complex food supply chains, as it involves the movement of multiple perishable food products across different stakeholders and complex processing operations. Henceforth, making it a challenging process to ensure food safety for the customer. Thankfully we have blockchain, a new-age technology that enables enterprises to create and manage traceable supply chains [4,5,6,7]. In its initial years, blockchain was limited in its applicability around the cryptocurrency domain. As time followed, researchers began to identify numerous advantages of integrating blockchain technology across different verticals. One of the recent areas of blockchain research involves its applicability in the area of supply chain management. Blockchain allows the creation of a decentralised supply chain with immutable transaction records. It helps in tracking the origin of a product and its raw materials from farm to fork [8]. The use of a blockchain-enabled dairy supply chain would allow multiple stakeholders such as farmers, consumers, government authorities, and shipping companies to be on one single platform [9,10,11]. Information regarding products and their movements will be shared with these stakeholders transparently and securely. Apart from providing traceability and transparency to the supply chain, blockchain ensures reduced operational costs and helps in automating the decision-making process. Blockchain technology allows a customer to be aware of the nature and quality of a product and its origin. Blockchain technology coupled with the Internet of Things (IoT) and Cloud has the potential to monitor critical parameters of a dairy product along the entire supply chain. The quality parameters of a dairy product can efficiently be monitored and communicated using a distributed ledger technology such as blockchain. Reduced transportation time, faster payment settlements, and prevention of unnecessary food wastage can be ensured by implementing a blockchain-enabled dairy supply chain. Being an emerging area of research, blockchain-enabled food supply chains have their share of impediments at people and process levels. Despite the challenges, we believe blockchain has the potential to transform the dairy supply chain and facilitate the establishment of trust and willingness to purchase among customers. The following are some of the prominent contributions of our work:Identifying existing problems in the Indian dairy industryAnalysing the benefits of blockchain integration for creating a dairy food traceability systemCreation of an end-to-end supply chain management platform for the dairy industryCreation of a blockchain-enabled food safety platform for maintaining the quality of dairy products and their nutritional value throughout the supply chainCreation of a platform that enables its customers to identify levels of adulteration and contamination in dairy productsPredicting sales of dairy products concerning product type, region, retailer, and distributorPreventing overproduction and minimizing wastage of dairy products by the use of machine learning and blockchain technology

The rest of the paper is organized as follows. Section 2 talks about the research questions and impact dimensions addressed by the paper. In Section 3, we discuss some of the recent works concerning the applicability of blockchain technology in food traceability along with the works describing the impact of adulteration and contamination on food quality and its nutritional values. The extent of the Indian dairy industry and standard practices for processing, storage, and circulation of dairy products by the Indian government are discussed in Section 4. In Section 5, we present the success factors for adopting blockchain technology as a game changer for supply chain management. Section 6 talks about the system architecture for the proposed blockchain platform along with its various actors. In Section 7, we describe in detail the smart contracts being implemented for the execution of transactions between different stakeholders of the supply chain. Section 8 presents some of the most prominent blockchain adoption challenges for creating a food traceability platform. Section 9 explains use cases pertaining to different impact dimensions for easing the practical implementation of the proposed food safety and traceability platform. Section 10 illustrates the experimental setup of the proposed blockchain platform. Finally, Section 11 summarizes our findings and concludes the work. The following Figure 1 depicts the entire structure of the paper.

## 2. Research Methodology

In line with the aim of the paper, a systematic literature review was conducted. The purpose of the literature review was to identify, analyse, and categorise recent works concerning the applicability of blockchain technology in the food and dairy supply chain. The comprehensive literature review assisted us in understanding the characteristics of blockchain technology in enabling the creation of food traceability systems. Research gaps were identified concerning food traceability and the dairy supply chain. Secondly, it enabled us in creating the following research questions which we address throughout different sections of our work.

RQ1:What features of blockchain technology can assist in the creation of a sustainable supply chain for the dairy industry?RQ2:What are the ways in which the use of blockchain-based smart contracts can facilitate the prevention and identification of contamination in dairy products?RQ3:What are the ways in which the use of blockchain-based smart contracts can facilitate the prevention and identification of adulteration in dairy products?RQ4:How can a blockchain-enabled dairy supply chain prevent socio-economic implications of contamination and adulteration of dairy products?RQ5:How can a blockchain-enabled dairy supply chain support the creation of a circular economy for the dairy industry?

The literature review was conducted for the time period of 2017–2022. Figure 2 describes the process followed during our research. The following are the sets of search terms used during our survey:Blockchain + Supply Chain;Blockchain + Food Supply Chain;Blockchain + Dairy Supply Chain.

The search covered titles, abstracts, and keywords for peer-reviewed publications from the Scopus database [12]. The distribution of publications over time and classification of publications were presented in the following section. Table 1 illustrates the comparative analysis between some of the recent works in areas of food and dairy supply chain. Table 2 and Table 3 are constructed subsequent to the literature review. Apart from defining research questions, the literature analysis helped us create areas of maximum impact, known as the impact dimensions. There are four impact dimensions which have been identified and each of them represents a specific area of influence. The following are the four impact dimensions:Impact Dimension 1:* Social*Impact Dimension 2:* Economic*Impact Dimension 3:* Operations*Impact Dimension 4:* Sustainability*

The concept of creating impact dimensions helped us in achieving a multidimensional research approach through our work. Every impact dimension is related to a certain set of research questions. The following Figure 3 illustrates the relationship between impact dimensions and research questions.

## 3. Literature Review

The paper aims to explore ways in which blockchain technology can prove beneficial in overcoming food safety issues in the dairy industry of Vietnam [25]. It examines how the people of Vietnam have been suffering from a crisis in food safety. The authors propose a food traceability framework that combines the use of blockchain technology and the Internet of Things. The paper focuses on milk as the product of choice and details its journey from the farm to the processing plant. All processing steps which the milk undergoes are recorded and uploaded onto the blockchain in an automated manner by the use of sensors. The use of QR codes can be widely seen in the work as it acts as a gateway to all information concerning a particular product. The proposed framework is successful in presenting a traceability system and allowing the customer to have complete authentic information regarding a product being purchased.

The authors propose a blockchain-enabled smart contract-based system for payment settlements in the supply chain [27]. The work explores the risks associated with organizations failing to withhold their financial commitments in an agreed period. The primary objective of the paper is to uphold payment options such as cash, credit, and advance without any hidden or intermediatory cost being associated. The proposed smart contract provides traceability of a product all along the supply chain. Being a blockchain-enabled solution, any transaction performed in the supply chain is prevented from any form of data tampering and misinterpretation. The prosed system aims to establish trust during processes relating to payments, inventory audits, and asset traceability.

The work proposes a traceability system that is low on energy consumption, and cost-saving with minimum impact on the environment [24]. Algorand is the blockchain network that is used by the authors for creating the blockchain. The authors use the Pure Proof-of-Stake as a consensus algorithm to build a traceability platform on the principles of green blockchain which requires minimum computational power and is scannable in nature. The case study discussed by the authors is of the dairy industry with “Fontina PDO” which is a semi-cooked cheese as the target product.

The work presents a dairy logistic ecosystem that is built using blockchain technology [23]. The authors use a combination of blockchain technology and IoT to track the temperature of dairy products across the supply chain. The proposed decentralized supply chain collects and stores real-time data collected by various sensors relating to milk as it is the target product. This research aims to ensure the safety of dairy products and reduce any operational errors during the process of milking, transportation, processing, and distribution. The system allowed stakeholders to store and share regulatory certifications in a tamper-proof manner over the blockchain.

The proposed work discusses a blockchain-enabled decentralized supply chain “NUTRIA” for the dairy industry of Switzerland [22]. A series of interviews were conducted by the authors to collect data concerning the expectations of people and different stakeholders in a dairy supply chain. The purpose of this research is to develop an application that allows traceability of milk from farm to shelf. Each product is associated with a QR code that stores information regarding the physical flow of the product across the supply chain. Ethereum is the blockchain platform used for creating the application and every transaction that occurs in the supply chain is recorded on two supply chains.

The authors propose a blockchain-enabled supply chain architecture that supports tamper-proof data sharing on a real-time basis [17]. Using the proposed architecture, the authors ensure creating an audit trail throughout the food supply chain. The paper is written in the backdrop of COVID-19 and ensures that the proposed architecture can trace all practices along the food supply chain and minimize the risks associated with bacteria, fungi, and parasites. Information such as animal breed, age, and temperature of employees working in processing plants are stored on the blockchain. The proposed work facilitates streamlining documentation and reducing unnecessary paper trails required by government authorities for importing food products. Tamper-proof quality certificates relating to food products are stored and circulated using blockchain transparently and authentically.

The work discusses the creation of a supply chain traceability system based on blockchain and RFID technology [19]. The proposed platform allows storage and management of data collected from RFID readers onto the blockchain. Hyperledger Fabric 2.0 was used as the development platform to create a consortium blockchain. The proposed architecture aims to track products across stages of shipment, stocking, and storage in the supply chain. The architecture comprises three parts, namely RFID tags, RFID reader, and blockchain platform. The RFID tag contains information related to the product and the hash value of the previous block. Every time an RDIF tag passes through an RFID reader, a new block is generated. The RFID reader is connected to the backend blockchain platform and shares with it all information scanned from the RFID tags. The end user is connected to the blockchain platform through a user interface for accessing information about a particular food product in the supply chain.

The authors propose a blockchain-enabled traceability system for extra virgin olive oil (EVOO) production on an Italian farm [18]. The authors believe that the quality of EVOO is directly linked to the cultivators being used and the environmental conditions of their growing area. The traceability system comprises thirty-three olive trees from three different cultivators. The purpose of the traceability system is to trace the origin of an EVOO product back to its olive tree. The prototype of the proposed system was implemented in the orchards of Italy. The information gathered by the traceability system assisted the farmers and manufacturers at an economic level.

Authors [28] reviewed milk adulteration, its reason, and its effects on human health. They reported that along with economic benefits the reason for milk adulteration is a high ratio of demand to supply. In many parts of the country, the milk delivered is not as per the FSSAI standards. There is a need for an effective and consistent quality check for monitoring the quality of milk and milk products. Keeping in mind the statement issued by WHO that to avoid serious diseases in the major population of India a legitimate checking of milk adulteration is required, the authors proposed that a human and technology interface, cognizance, and traceability are the need of the hour for adulterant-free milk for consumption.

Authors defined contamination issues in dairy products as a global problem [10]. They reviewed several scientific literature and other databases from 2015 to 2019 and found that not only biological contamination but physical and chemical contamination are serious concerns for human health. The authors extracted safety and fraud data of milk and milk products from online resources such as HorizonSCan and the EU Rapid Alert System for Food and Feed (RASFF). The data reflected that most cited microbial contamination or malpractices in production were for cheese. The work also mentioned that there are many reports of contamination and adulteration as grey literature. It is vital to be aware of the safety, legitimacy, and quality of milk and products by all the stakeholders.

Authors [29] said that the adulterants are added to milk for financial gain but the contaminations are due to a lack of knowledge of the processing of milk, hygiene consciousness, and insensitivity to human health risks. Water, synthetic milk, sugar, and benzoic acid are the most common adulterants used in milk. Benzoic acid is added to increase milk’s shelf life, and its addition can cause adverse effects on consumers.

Adulteration in milk and products is causing serious impacts on human beings including gastrointestinal and cardiac issues, endocrinal imbalances, neural disorders, etc. [30]. For instance, starch, a common adulterant, induces diarrhoea as its intestinal absorption is difficult. Starch can also shoot up sugar levels and can be lethal for diabetic patients. The evaluation of milk properties, qualitative as well as quantitative, is a must and proper traceability is desirable to avoid all harmful impacts on human health and the nutritional value of milk and products. The authors described different methods for the detection of contaminants and adulterants in milk products such as pulverized soap, benzoic acid, skim milk powder, sugar, Vanaspati, salicylic acid, starch, etc.

## 4. Overview of the Indian Dairy Industry

India has been one of the largest producers and consumers of dairy products across the world since the late 1990s. The dairy industry is an essential part of the Indian economy and has a significant role in generating rural employment [31,32,33,34]. As of 2021, the Indian dairy market reached a valuation of INR 13,174 billion with a growth trajectory of reaching INR 30,840 billion by the year 2027. Most of the dairy products in India are consumed domestically, with milk having the largest consumption value. In terms of milk production, India produced more than 198 million tonnes of milk in the year 2019–2020. As per a study conducted by the National Dairy Development Board (NDDB), India will be producing 266.5 million metric tonnes of milk by 2023. States such as Uttar Pradesh, Rajasthan, Madhya Pradesh, Gujarat, and Andhra Pradesh are considered to be the highest producers of milk. In India, the dairy sector holds significant importance as it has huge implications for the socio-economic aspects of its people. Keeping this in mind, the Government of India has launched the National Dairy Programme to boost cattle productivity, enhance milk production, and therefore improve the livelihoods of the farmers. Strategic investments are being made by the governments to improve milk procurement infrastructure in rural parts of India. In recent years, the Indian dairy sector has seen a significant rise in the production and consumption of milk-related value-added products (VAP). Products such as cheese, butter, and yoghurt are witnessing a rise in per-capita consumption. Despite numerous government schemes and the largest bovine population, the milk production per animal in India is less than its contemporaries such as the US and UK. The use of non-scientific methods, inefficient cattle breeding, poor management strategies, and above all the lack of technology intervention are some of the reasons that have led to the diminished growth of the Indian dairy sector. Moreover, a majority of the Indian dairy sector is unorganized in nature and therefore unable to adapt to new technologies and reach competitive markets [35,36,37,38,39]. With 3/4th of the sector being unorganised, the Indian dairy sector has emerged as a breeding ground for numerous malpractices. The absence of strict audit mechanisms and lack of quality certifications from the governments have led to the surplus use of contaminated and adulterated dairy products. Circulation of adulterated dairy products has become the new normal for the Indian dairy sector causing severe health implications for its people. It is believed that 79% of milk available in the Indian market is adulterated to a report presented by the Consumer Guidance Society of India (CGSI) in the year 2020. The food supply chains in India are significantly unorganised and perform worse when dealing with perishable food items such as dairy products. The Indian dairy supply chain is highly fragmented and even dysfunctional at certain levels. Poor linkages between different stakeholders within the supply chain have led to a shortage of dairy products and even caused price inflation. The Indian dairy supply chain is highly dependent on manual handling thus resulting in hygiene issues and human errors. The supply chains lack the presence of regulatory authorities at small and medium scales thereby allowing the entry of adulterated dairy products into the supply chain. The Indian dairy supply chains are mostly devoid of technologies and therefore are unable to address the changing consumer behaviours and global export initiatives.

A food product must be safe to consume and all the stakeholders (farmers, consumers, cooperatives, processors, and government agencies) share a responsibility to ensure the suitability of a food item. In the dairy industry, regulatory laws are in existence since 1899 for the safety and quality of milk and its products. There have been substantial amendments and increments in the legislation and quality standards of milk and products. The Food Safety and Standards Act received a nod in the year 2006.

In 2008, the Food Safety and Standards Authority of India (FSSAI) was established under the same act. The FSSAI enforced a regulation to control adulteration and contrived milk products known as Food Safety and Standard Regulations 2011. Under the regulation, Section 2 mentions the permissible limits of contaminants, toxins, and residues in milk and milk products. The Milk and Milk Products Order (MMPO) 1992 is regulatory order of the Government of India, under the Essential Commodities Act 1955, for rheostat of production, supply, and distribution of milk and milk products to uphold or enhance the supply of milk and products from producer to consumer. The MMPO is to rationalize legal registration, define the terms used for milk, e.g., boiled milk, pasteurized milk, etc., and control restrictions on irrational procurement and supply of milk, stringent rates of production, sincere hygiene environment, proper packaging, labelling, penalization, etc. Thus, MMPO warrants the safety and progress of the Indian dairy industry. The Indian dairy supply chain can benefit significantly from the implementation of blockchain technology. Functionalities such as stakeholder management, inventory management, product movements, and returns management can easily be achieved through the use of blockchain technology. Blockchain-enabled dairy supply chains would ensure last-mile product delivery, quality control mechanisms, inventory optimizations, fault analysis, and pricing optimizations [40]. Blockchain technology presents itself as a solution to the numerous limitations of the Indian dairy supply chain [41,42,43,44]. A detailed description of some of the key factors enabling blockchain technology to be a perfect match for the dairy supply chain is presented in the following section.

## 5. Blockchain Success Factors for Dairy Supply Chain Management

Blockchain has proven itself to be a game-changer technology in the area of supply chain management. The presence of blockchain technology introduces aspects of transparency, trust and decentralization into the supply chains. The following Table 4 lists some of the prominent success factors leading to the adoption of blockchain technology for dairy supply chain management.

## 6. System Architecture

The significance of blockchain technology in the dairy supply chain has been discussed in the previous sections. We are aware of the problems being faced by the Indian dairy sector and how revolutionizing the supply chain holds the key to its transformation. In this section, we discuss the system architecture for our proposed supply chain model. The system architecture comprises four layers, namely the phases layer, traceability layer, blockchain layer, and application layer. All operations within the supply chain are performed using these four layers. The following sub-sections define each layer, their functionalities, and the ways in which they interact with one another to create a comprehensive sustainable dairy supply chain. Figure 4 presents a detailed overview of the system architecture.

### 6.1. Phases Layer

The phases layer talks about the various touch points a product passes through in an entire supply chain. It can also be represented as a collection of different stages of the supply chain. The first phase is the dairy farm where cows are milked every day and raw milk is collected by the farmers to be subsequently sold to the dairy company. The term “farmer” here refers to an individual farmer or a collection of farmers in the form of a cooperative society. Traceability starts at this very stage onwards as QR codes are used to store information pertaining to farmer details, breed of the cow, and timestamp of the milking process. All this data is updated onto the blockchain and thus cannot be tampered with by any individual. The next phase we have is the collection centre wherein milk from different dairy farms is collected and tested. It is the first phase where testing takes place for checking contamination and adulterations in milk collected. A smart contract is executed between the farmer and dairy company allowing the purchase of raw milk. Raw milk which fails to meet the required quality standards is deemed unfit for purchase and returned to the farmer. Post-purchase completion, milk is packed in batches and assigned respective QR codes. After the collection centre, the milk is further transported to the processing centre. The QR codes mentioned on the milk batches comprise information regarding the transportation vehicle, source, destination, and timestamp. The transportation phase is responsible for the movement of milk and its related products, in our case cheese and butter, across different stakeholders in the supply chain. The transportation phases update information on the respective QR codes for individual products and different product batches. Various shipments moving across different locations through different modes of transportation are tracked and verified using these QR codes. It is to be noted that the system facilitates minimum paper trail and ensures a single document per shipment across various modes of transportation. A smart contract is executed between the dairy company and the transportation firm for automated settlement of payments. Milk from various collection centres is brought to processing and packaging centres for processing raw milk and converting them into different products such as cheese and butter. Multiple quality tests pertaining to contamination and adulterations are performed at the processing and packaging centre. Any adulterated or contaminated batch is strictly discarded and removed from the supply chain. New QR codes are generated for the packed products having updated information regarding the nature and quality of the product. Processed milk products are then further sent to either cold storage centres or different distributors and retailers. Random quality checks are performed at the distributor and retailer level to prevent the selling of any contaminated or adulterated dairy product. All transactions performed between the different phases of the supply chain are recorded, validated, and stored on the blockchain. Figure 5 explains the various levels of interaction between different stakeholders in the phases layer.

### 6.2. Traceability Layer

Traceability is one of the key aspects of any modern-day supply chain. The traceability layer of the system architecture tracks the movement of every raw material and dairy product throughout the supply chain. The movement of a product from one stakeholder to another is marked as a transaction and recorded on the blockchain. Being a blockchain-enabled solution, the traceability information is tamper-proof and shared among all stakeholders. Information relating to a particular product, process, stakeholder, and transportation is collected and managed using the traceability layer. Information pertaining to any form of contamination or adulteration that has occurred in a particular product or group of products at any stage of the supply chain is identified and shared using the traceability layer. From an organization’s point of view, sales information holds significant importance as it is directly related to its revenue streams. The traceability layer tracks and manages information regarding sales of individual products or a type of product. The layer also supports accountability as it only allows authorized entities to perform a transaction, i.e., recognized stakeholders are only permitted to transact a dairy product and update the information regarding its nature and movement. Information concerning any fake product cannot be entered into the system thereby preventing entry of any counterfeit product into the supply chain. It is through the execution of smart contracts which allows all the above information to be exchanged between stakeholders.

### 6.3. Blockchain Layer

The foundational layer of the entire system is the blockchain layer. It is the layer which provides all channels of communication between different stakeholders of the supply chain. Multiple stakeholders first need to register themselves on the blockchain to conduct transactions and subsequently verify them. Ethereum is the blockchain network that has been used for creating the supply chain model. Solidity-based smart contracts are used to ensure secure and automated transactions between different stakeholders. A collection of transactions is called a block. Each block in the blockchain has a link to its predecessor block. Being a distributed ledger technology, blockchain allows sharing of information transparently and securely. The presence of blockchain technology enables the supply chain to be completely decentralized and prevents any single point of failure. Transactions such as shipping, billing, quality check, and adding products are executed on the blockchain using smart contracts. The blockchain layer ensures immutability in the supply chain thus once a transaction has been validated and stored it cannot be reverted. The blockchain layer establishes connectivity between other layers of the system architecture. All information including information collected from the transactions performed by the phases layer to information relating to the traceability layer is stored and shared using the blockchain layer.

### 6.4. Application Layer

The application layer is the one which directly relates to the retail customers. The utility of the system is experienced by the end user through its interaction with the application layer. The overall supply chain management system is presented to the end customer employing the application layer. The application layer comprises functionalities that have their roots present in the above three layers. All functionalities being rendered by the application layer are derived from individual traits of the previous three layers. Quality management is one of the most important aspects of the application layer. It allows the customer to be aware of the nature and extent of contamination or adulteration that a particular food product has encountered during its movement in the supply chain. Quality management assures the prevention of any contaminated or adulterated product to reach the customer. It also prevents the entry of fake products within the supply chain. The quality management aspect is also helpful for the dairy company as it allows the company officials to identify the source of contamination or adulteration. It enables the officials to identify the entry of a counterfeit product and trace it back to its origin, i.e., the distributor or retailer through which it was introduced into the supply chain. Contamination and adulteration checks are performed at every level of the supply chain at both product and batch levels. Besides assessing the impact of contamination or adulteration on a product, assessing its nutritional value also holds significant importance. Dairy products such as milk, cheese, and butter are seen as primary sources of nutrition and are consumed across different age groups. Henceforth, it becomes even more essential to maintain the nutritional value of a product throughout the supply chain. The quality management aspect ensures nutritional values of a product are checked and updated at every stage of the supply chain. The QR code associated with a product holds this updated information concerning its nutritional values. Customers will be able to access these values by a simple scan of the QR code. Tracking changes in the nutritional values of a product can be a valuable insight for the dairy company as it will facilitate them to make changes in their processing mechanisms and supply chain routes. All these efforts will be allied to ensure the nutritional values of a product remain intact throughout the supply chain.

Sales management is another important aspect of the application layer. It allows the dairy company to keep track of the type and quantity of the products being sold to a particular distributor and retailer over a specific time. Sales management provides valuable insights concerning the sales patterns of different products across different regions at different time intervals. Companies can launch new products, withdraw existing ones, and make changes to their sales and marketing strategies all based on these sales patterns. Moreover, the companies can vary their production rates for different products seeing to their sales patterns and acceptability among customers. A similar fate can be achieved for products showing signs of depleting nutritional values and high perishability. Being a blockchain-enabled solution, the application layer comprises an aspect concerning the management of payments and settlements. The problems of delayed payments, irregularities in payments, and missing payments are completely prevented by the application layer. Financial settlements happen automatically between stakeholders by the execution of smart contracts.

## 7. Smart Contracts

Smart contracts are an integral part of any blockchain-based solution. The section describes in detail the list of all smart contracts implemented in our work. Table 5 talks through the various functions incorporated by different smart contracts. Figure 6 describes the execution flow followed by the smart contracts. Appendix B lists all the smart contracts (Appendix A) that were created and deployed for ensuring the functioning of the proposed traceability platform.

## 8. Blockchain Adoption Challenges

The following Table 6 describes some of the leading adoption challenges being faced by blockchain technology with respect to its implementation in the dairy supply chain.

## 9. Use Cases for the Dairy Supply Chain

In this section, we will be illustrating the implementation benefits of the proposed blockchain-enabled food safety and traceability platform by means of use cases. Each use case is a replica of a real-world scenario that a dairy company or an end customer may encounter during the production, storage, sales, and purchase of dairy products. The use cases aim to enable the readers in having a better understanding of the proposed platform and enhancing their sense of correlation between different processes of food safety and traceability and stakeholders of the supply chain. The use cases are aligned with the impact dimensions and comprise individual objectives. In total, there are five use cases, one each for economic, operations, and sustainability impact dimensions and two use cases concerning the social impact dimension. Appendix A lists Table A1, Table A2, Table A3, Table A4, Table A5 and Table A6 which represent the impact of contamination and adulteration on all three dairy products.

### 9.1. Social


**Use Case 1**
**Dairy Products:** Milk, Cheese, and Butter
**Objectives:**
Identification of AdulterationPrevent Nutrition DepletionPrevent the occurrence of disease as a result of product consumption


**Description:** A dairy company named Parag Enterprises is operating in the states of Uttar Pradesh and Uttarakhand, India. Milk is its primary selling product apart from cheese and butter. The company has more than 200 points of sales across the two states. It currently has 10 distributors and each distributor is associated with 20 retailers. Since the past couple of weeks, reports of diarrhoea have been reported by customers consuming its milk products across the regions of Dehradun and Haridwar. Multiple complaints were being reported to its distributors concerning stomach infections and diarrhoea thus resulting in bad publicity. Implementing the proposed blockchain-enabled supply chain platform assisted the company to identify the entry of fake milk products into the supply chain. The platform supports adulteration checks at different stages of the supply chain and identified fake milk packets comprising starch and urea leading to customers suffering from gastro disorders and diarrhoea. Using the platform, the company was even able to identify the particular retailer responsible for selling fake milk packets using the company’s logo. In months to follow, customers in the region of Srinagar, Uttarakhand raised complaints regarding certain cheese products of the company. Similar complaints were being registered by customers from Dehradun regarding the consumption of cheese products. Using the food traceability platform, a set of retailers, distributors, and processing centres were identified for the adulterated products. Quality tests were reconducted at all three levels leading to the identification of a processing centre responsible for adding palm oil during the process of cheese production. Henceforth, causing all operations to be suspended at the processing centre and dairy products produced to be recalled. The proposed platform proved to be an excellent aid in tracking all contaminated cheese products spread across different cities and removing them from the supply chain all at once. Thereby, ensuring the safety of customers from serious cardiac problems.


**Use Case 2**
**Dairy Products:** Milk, Cheese, and Butter
**Objectives:**
Identification of ContaminationPrevent Nutrition DepletionPrevent the occurrence of disease as a result of product consumption


**Description:** A dairy company named Gopal Enterprises is operating in the states of Uttar Pradesh and Uttarakhand, India. Milk is its primary selling product apart from cheese and butter. The company has more than 200 points of sales across the two states. Recently, the company’s processing centre has detected batches of raw milk being unfit for consumption and further processing. After performing a series of quality tests, mycotoxins were found in different batches of milk. Implementing our proposed blockchain-enabled supply chain platform assisted the company to update information regarding contaminated batches of milk and remove them from further entering the supply chain. Moreover, the platform helped them in tracing back to the collection centres and dairy farms associated with the contaminated batches of milk. Subsequently, the dairy farms were removed as stakeholders from the supply chain. A similar case of milk contamination was detected with the help of the blockchain-enabled supply chain platform at a Kanpur-based distributor of the company. Cases of decreased shelf life for milk packets were being reported by retailers in parts of Kanpur and Uttar Pradesh. As a result of phase-wise testing supported by the platform, company authorities were able to trace and detect microbial presence in packets of milk at the distributor level. All contaminated milk packets were subsequently removed from the supply chain. In months to come, complaints started surfacing regarding products such as cheese and butter. The distributors of Dehradun, Uttarakhand reported cases of reduced shelf life for certain batches of cheese and butter. Within no time of implementing the proposed blockchain-enabled platform, company authorities were able to find traces of organic pollutants such as dioxins in certain samples of cheese and butter. The traceability platform enabled the authorities to trace the provenance of these products along with identifying the other intermediatory stakeholders. Soon it was noticed that certain cold storage centres in Saharanpur, Uttar Pradesh were responsible for such levels of contamination. The authorities quickly reacted and removed all batches of cheese and butter that were shipped from the concerned cold storage centres. The proposed food safety and traceability platform facilitated the speedy removal of contaminated products and prevented occurrences of diseases such as diarrhoea and cancer among customers with frequent consumption of the contaminated products.

### 9.2. Economic


**Dairy Products:** Milk, Cheese, and Butter
**Objectives:**
Identify Sales PatternsPredict Future SalesIncrease in RevenuePrevent Food Wastage



**Description:** A dairy company named Mohan Enterprises is operating in the states of Uttar Pradesh, Uttarakhand, and Madhya Pradesh, India. The company is a renowned cheese producer in the northern region of India. Apart from cheese being its primary selling product, the company also has production lines for milk and butter. The company has more than 500 points of sales spread across the three states. The state of Uttar Pradesh has a maximum of 20 distributors with each distributor being associated with 15 retailers. The state of Rajasthan has 15 distributors with each distributor being associated with 10 retailers. Uttarakhand has 5 distributors with each distributor being associated with 10 retailers. The company plans to come up with a growth strategy that allows it to launch new products and target new customers. In view of its expansion plans, the company incorporated the proposed blockchain-enabled supply chain platform. The platform enabled the company to acquire sales data for every individual product at a distributor and retailer level. The platform allowed the company to possess geo-specific sales data for individual products. Moreso, the company was able to perform data analytics on the mentioned data using machine learning algorithms. The algorithms created trend lines for sales concerning products, locations, distributors, and retailers. Using the platform, the company was able to identify purchasing patterns among customer groups and establish a correlation between sales of different dairy products. Specific product sales patterns were generated with respect to different months of the year. Collective knowledge acquired from the above data analytics enabled the company to increase the sales of its current products and formulate a sales strategy for its future products. Apart from direct benefits to the company’s revenue, the blockchain platform proves its utility by preventing the wastage of food. Dairy products being highly perishable in nature have a limited shelf life thereby making it necessary for the company to produce them in an optimised manner. The blockchain platform allows the company to establish a trade-off between product sales and its shelf life. The platform enables the company to maintain optimised production levels of its dairy products while adhering to sales patterns and perishability, henceforth, preventing food wastage.

### 9.3. Operational

**Dairy Products:** Milk, Cheese, and Butter
**Objectives:**
Quality Assurance CertificateAutomated Payment SettlementsTraceabilityDecentralizedAccountabilityMinimum Documentation Trail


**Description:** A dairy company named Krishna Enterprises is operating in the states of Gujrat and Rajasthan, India. Milk is its primary selling product apart from cheese and butter. The company has more than 400 points of sales across the two states. The state of Gujrat has 10 distributors with each distributor being associated with 30 retailers. The state of Rajasthan has 10 distributors with each distributor being associated with 10 retailers. A majority of its collection and processing centres are located in the state of Gujrat, whereas the cold storage centres are predominantly present in the state of Rajasthan. Recently, the company has been experiencing significant extents of bad publicity from its local competitors. In response to the situation, the company adopted the proposed blockchain-enabled food safety and traceability platform. The platform assisted the company to release quality certificates for its products by government standards. The quality certificates were accessible to end customers by scanning the QR code present on the product. In months to come, the company started to face issues regarding payment settlements from a few of its retailers. Few distributors from Rajasthan had rendered credit more than the acceptable limits to their retailers thereby resulting in a financial irregularity. The company management was unaware of such illicit practices as they only surfaced after an internal audit. The practices were going on for months resulting in an imbalance in the accounting books. With the implementation of the proposed blockchain-enabled supply chain platform, the company was able to ensure a culture of real-time payment settlements as soon as the products were delivered to the retailers. The platform allowed the company to monitor payments at a stakeholder level. A single document was generated for every order by a distributor or retailer comprising all shipment and product details. The document was timestamped every time a product passes through a stakeholder in the supply chain. Being a decentralized platform, no single authority possesses governance rights over the document thus proving it to be tamper-proof and ensuring its authenticity. Moreover, the company was able to identify the set of distributors and retailers who were responsible for multiple financial irregularities at different time intervals. The platform ensured accountability of products being sold by every distributor and retailer. Any instance of a counterfeit product was timely identified and subsequently removed from the supply chain.

### 9.4. Sustainability

**Dairy Products:** Milk, Cheese, and Butter
**Objectives:**
Reuse of Dairy Products (Circular Supply Chain)Prevent Wastage of Dairy ProductsPrevent Shortage of Dairy ProductsEfficient disposal of Contaminated and Adulterated Dairy Products


**Description:** A dairy company named Sohan Enterprises is operating in the states of Uttar Pradesh and Bihar, India. The company is a renowned butter producer in the northern region of India. Apart from butter being its primary selling product, the company also has production lines for milk and cheese. The company has more than 250 points of sales spread across the two states. The state of Uttar Pradesh has a maximum of 15 distributors with each distributor being associated with 10 retailers. The state of Bihar has 10 distributors with each distributor being associated with 15 retailers. Recently, a quality check in one of the processing centres of Lucknow district traced contaminated milk. The contamination made milk unfit for direct consumption. Implementing our proposed blockchain-enabled supply chain platform the company gets updated regarding contaminated batches of milk. As a result of phase-wise testing supported by the platform, company authorities were able to trace and detect contamination in packets of milk at the distributor level. All contaminated milk packets were subsequently collected at the retailer, distributor, and processing centre level. The timely detection ensured more release of milk products from the company, preventing a shortage of milk in the market. The milk was tested for kind of contamination and it was found to be microbial. The batch number and order number helped to predict the shelf life of pasteurized milk. It was found that 25% of the milk can be converted to some other milk products as the extent of spoilage is not too severe and milk can be reused after processing. The dairy company converted the fluid milk into other dairy products such as cheese. However, it was found that the rest of the milk can only be thrown away as the impact of contamination was severe, making fluid milk not suitable for reuse. Our experiment has ensured whether to use spoiled milk or discard it. Our platform helped in the proper handling and sustainable disposal of spoiled milk. The company made proper arrangements to discard the milk and prevented the accumulation in landfills that may produce methane gas, pollution in water, and increased chemical oxygen demand. In another incidence of a random quality check of a sample of butter at one of the distributors of Patna, it was found that the sample was not up to the mark. The final analysis showed that it was adulterated with starch. The blockchain-enabled supply chain helped to trace all the batches adulterated with starch. Being the leading distributor of butter, the supplier ensures the makeup of the shortage of butter in the market. It also helped to collect all the samples of adulterated butter for proper disposal as per the government norms and sustainability parameters.

## 10. Experimentation Setup

The proposed dairy food safety and traceability platform are simulated using the Ethereum blockchain [46]. The smart contracts mentioned in Section 7 are developed using the Solidity language. All smart contracts are executed over the Ethereum test network. MetaMask has used a token wallet for stakeholders to make payment settlements using Ether. For depicting payment settlements, ethers are transferred from the wallet address of the buyer to the seller. VS Code and Remix are the two integrated development environments (IDEs) used for developing the entire blockchain-enabled supply chain platform. MetaMask acts as a bridge between Ganache and the Remix IDE. Python is used as a backend language for developing the decentralized supply chain platform. The QR codes comprising various information concerning a particular dairy product, batch, and order are generated using a python program. The end-to-end food safety and traceability platform is a conglomerate of multiple new-age technologies. We use Ethereum blockchain as a foundational technology coupled with Azure Cloud, IoT, and machine learning. Every technology implemented has a definitive role to play such as blockchain creating the underlying network, IoT supporting the collection of tracking data, cloud computing being used as secondary storage to the blockchain network, and machine learning algorithms used for prediction purposes. The seamless integration of these technologies and their orchestrated functioning results in the creation of our blockchain-enabled food safety and traceability platform. The following Figure 7 depicts various levels of gas consumption (wei) for different smart contracts.

## 11. Conclusions and Future Work

The paper proposes a blockchain-enabled supply chain platform for the dairy industry. The proposed platform ensures the safety and traceability of dairy products through the supply chain thereby preventing its customers from consuming counterfeit products. The research proposed was accomplished by addressing all research questions listed at the start of the paper. The paper considered milk, cheese, and butter as its target products and implemented a blockchain-based solution given their production, storage, and transportation processes. The Indian dairy industry was considered a test bed for our proposed platform. The scale and diversity of the Indian market along with its extensiveness in consumption of contaminated and adulterated dairy products make it the perfect choice for the implementation of our dairy food safety and traceability platform. The work illustrates a combination of technologies such as blockchain, IoT, Cloud computing, and machine learning working together in identifying and removing contaminated and adulterated dairy products from a supply chain. The work considers technology as an enabler for socio-economic change and presents its applicability across four impact dimensions: social, economic, operations, and sustainability. The social dimension deals with the health and wellbeing of dairy product consumers along with the nutritional benefits a dairy product has to offer. Thereby, ensuring the movement of contamination and adulteration free throughout the supply chain. The economic dimension is for ensuring the profitability of being part of the dairy industry at the levels of a dairy farmer as well as a large dairy enterprise. Future sales prediction and launching of new products are a part of the economic dimension. The operational dimension deals with the overall functioning of the supply chain. The work proposed ensures the creation of a decentralized, robust, and transparent supply chain. In terms of the operational dimension, the platform facilitates the establishment of trust among unknown entities and simplifies the process of resolving disputes among various actors in the supply chain. The privacy of every actor is preserved within the supply chain. Furthermore, the use of smart contracts eliminates unnecessary intermediaries thereby reducing the overall cost of managing the supply chain. Finally, we have the sustainability dimension concerning aspects of a circular supply chain. The work provides mechanisms for reusing discarded dairy products which are in line with the larger goal of fighting hunger. Table 1 presents a comparative analysis between our proposed platform and its contemporaries. The work was able to address the listed research questions but still comprises certain limitations to be addressed by future researchers. The proposed food traceability platform needs to be implemented on a large scale with real-time complexities to truly validate its functioning. Interlinking multiple dairy supply chains operating in different regions can be one of the many future research areas. The work primarily focuses on the Indian dairy sector but future researchers can extend its applicability to dairy supply chains ranging across different countries. Moreover, new consensus mechanisms can be proposed by researchers to ensure faster and energy-efficient transaction validations. Government authorities and leading private players can come on board with the platform and ensure the creation of nationwide food safety and traceability platforms.

## Figures and Tables

**Figure 1 foods-11-02716-f001:**
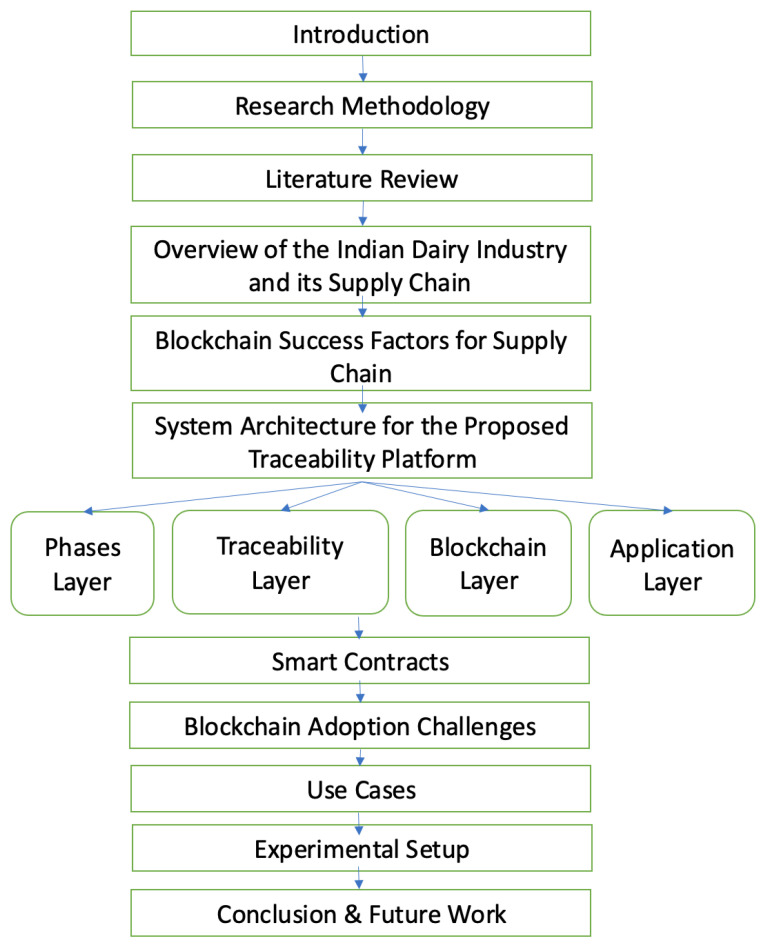
Paper Organization.

**Figure 2 foods-11-02716-f002:**
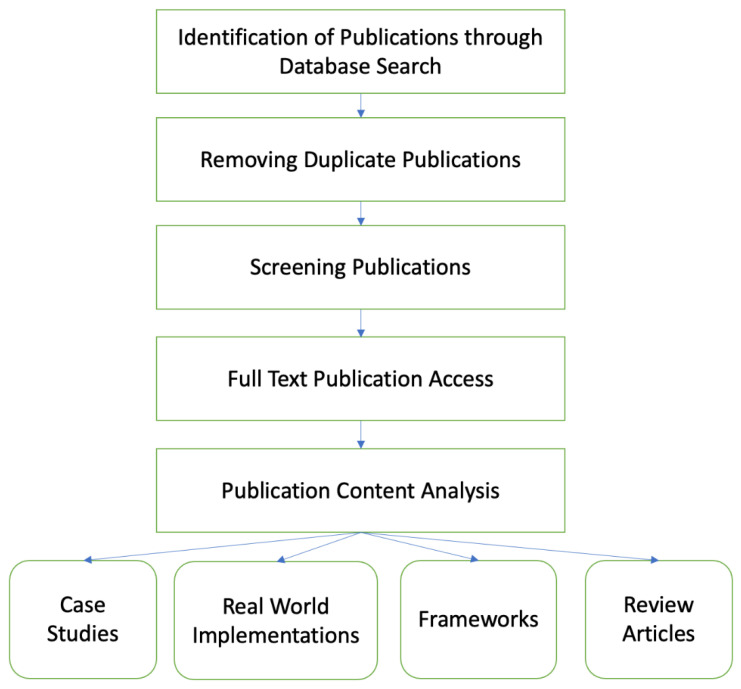
Research Methodology.

**Figure 3 foods-11-02716-f003:**
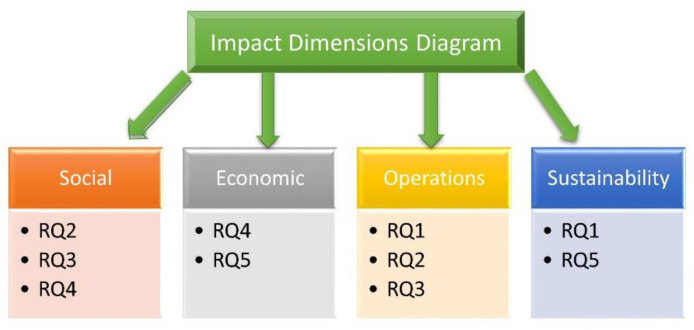
Relationship between Impact Dimensions and Research Questions.

**Figure 4 foods-11-02716-f004:**
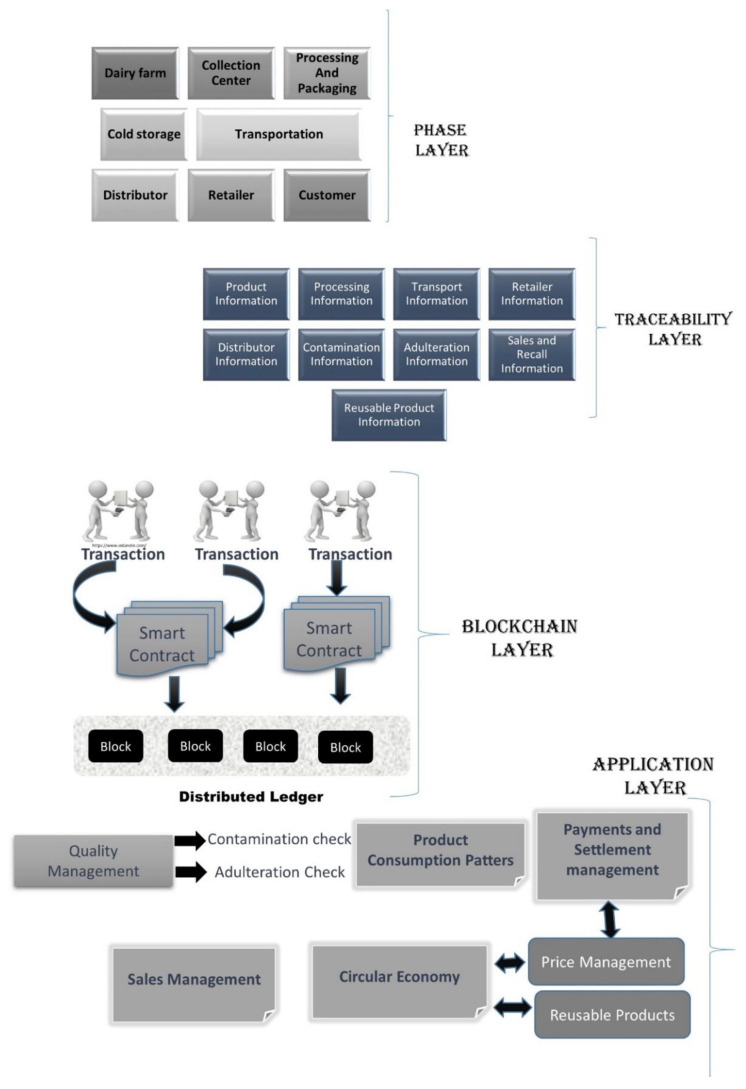
System architecture for the proposed blockchain-enabled supply chain platform for food safety and traceability.

**Figure 5 foods-11-02716-f005:**
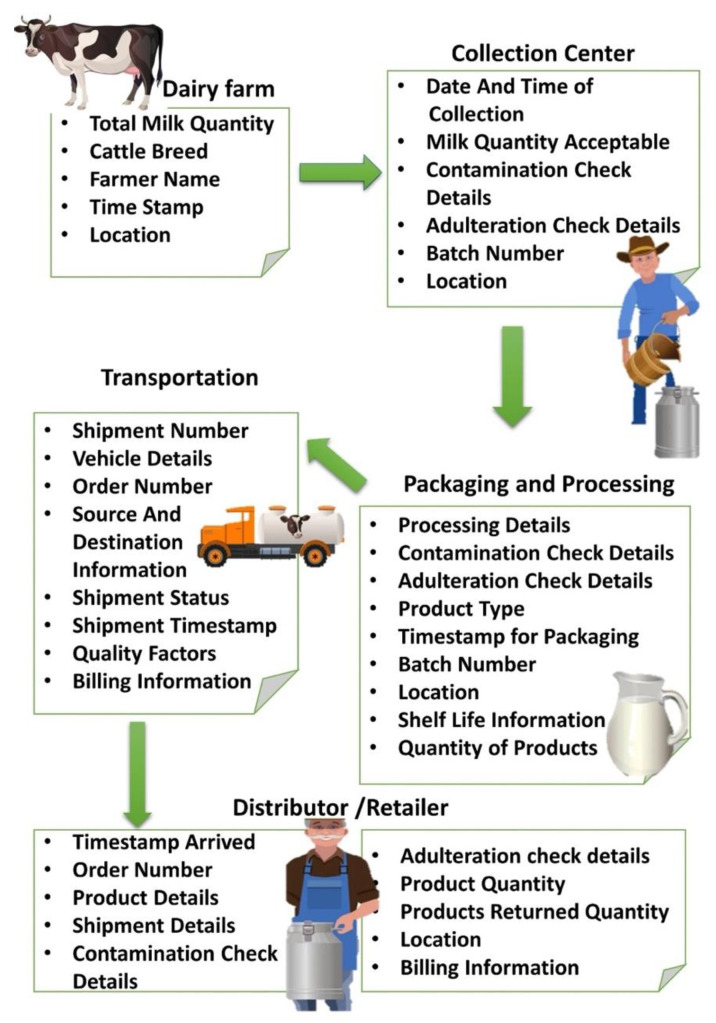
Information exchange within the food safety and traceability platform.

**Figure 6 foods-11-02716-f006:**
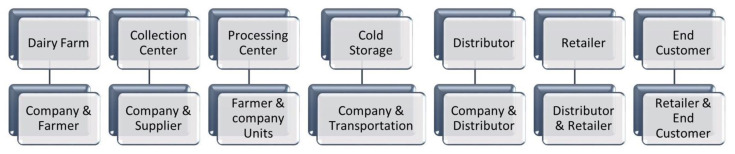
Smart Contract Execution Flow.

**Figure 7 foods-11-02716-f007:**
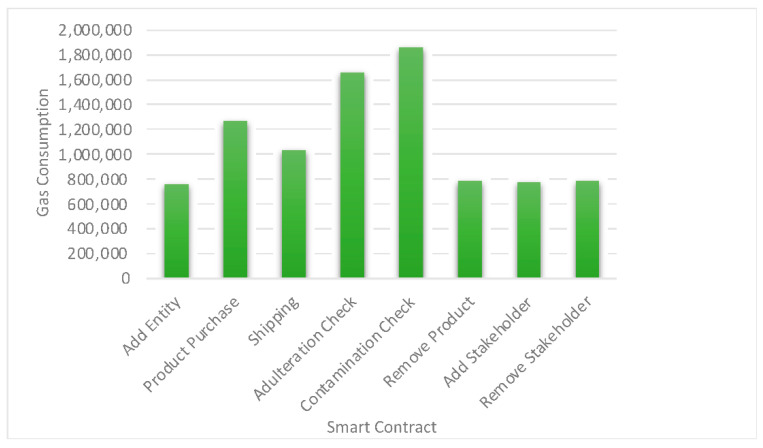
Gas Consumptions specific to smart contract execution.

**Table 1 foods-11-02716-t001:** Comparative analysis of recent works in food traceability and safety.

Publication	Year	Key Supply Chain Functionalities	Other Technologies Being Used	Sector	Dairy Products Being Catered	Related Impact Dimension
[13]	2022	Traceability, Sustainability	IoT	Beef Industry	NA	Operations, Sustainability
[14]	2022	Food Quality	IoT	Rice Crop	NA	Operations, Social
[15]	2022	Traceability, Food Quality, Food Shortage Detection	IoT	Food Industry	NA	Operations, Social
[16]	2022	Traceability	Machine Learning	Rice Crop	NA	Operations
[17]	2021	Traceability, Auditability, Decentralized	IoT	Grains	NA	Operations, Economic
[18]	2020	Traceability, Authenticity, Food Quality	Radio-frequency identification (RFID)	Extra Virgin Olive Oil	NA	Operations, Economic
[19]	2022	Traceability	RFID	Food Industry	NA	Operations
[20]	2021	Traceability, Authenticity	IoT, RFID	Food Industry	NA	Operations, Social
[21]	2021	Traceability, Authenticity, Food Quality, Decentralized	-	Dairy Industry	Milk	Operations, Economic, Social
[22]	2021	Traceability, Authenticity	-	Dairy Industry	Milk	Operations, Social
[23]	2021	Traceability, Decentralized, Transparency	IoT	Dairy Industry	Milk	Operations, Economic
[24]	2022	Traceability, Decentralized	-	Dairy Industry	Cheese	Operations, Sustainability
[25]	2020	Traceability	-	Dairy Industry	Milk	Operations
[26]	2020	Traceability, Food Quality	IoT	Food Industry	NA	Operations, Social
Our Proposed Model	2022	Traceability, Food Quality, Accountability, Product Authenticity, Automated Payment Settlements, Decentralized, Nutritional Value Assurance	IoT, QR codes, Machine Learning, Cloud Computing	Dairy Industry	Milk, Cheese, Butter	Social, Economic, Operations, Sustainability

**Table 2 foods-11-02716-t002:** Year specific publication analysis.

Year	Supply Chain	Food Supply Chain	Dairy Supply Chain
2022	566	105	4
2021	1091	180	5
2020	690	114	11
2019	465	68	3
2018	181	23	0
2017	45	4	0

**Table 3 foods-11-02716-t003:** Comparative analysis based on publication classification.

Document Type	Supply Chain	Food Supply Chain	Dairy Supply Chain
Article	1270	211	11
Conference Paper	1219	181	8
Book Chapter	181	32	2
Review	181	45	0
Conference Review	143	19	2
Book	13	0	0
Short Survey	8	3	0

**Table 4 foods-11-02716-t004:** Description of Blockchain Success Factors.

Success Factor	Description
Traceability	Blockchain facilitates a stakeholder to track the movement of a particular dairy product across the entire supply chain. The presence of a particular dairy product or a batch or a complete order can be traced by every stakeholder.
Transparency	All stakeholders within the supply chain are aware of any transaction being performed [45]. Information regarding a particular dairy product or a complete order is accessible to all stakeholders without any partiality.
Trust	Blockchain enables the establishment of trust between different stakeholders in the supply chain. Most importantly, it prevents the occurrence of a trust deficit between the end consumer and the dairy company.
Knowledge Sharing	Blockchain can assist in sharing valuable insights regarding the distribution and sales of a particular dairy product across different distributors and retailers. The important aspect is the safe and secure mechanism of sharing knowledge.
Smart Contracts	The most essential aspect of any blockchain solution is the use of smart contracts. They enable seamless transactions between stakeholders. Purchasing a product, managing multi-modal shipments, and removing a product are some of the many uses of smart contracts.
Tokens	Financial settlements can be made possible by the use of cryptocurrency tokens. Apart from methods such as cash and credit, tokens are more flexible, secure, and fast when it comes to handling payment settlements.
Immutability	The data stored on the blockchain is immutable in nature, i.e., once created it cannot be edited. A transaction once completed between two stakeholders cannot be revoked.
Auditable	Unlike traditional dairy supply chains, blockchain-enabled supply chains are auditable as every transaction performed within the blockchain network is recorded and stored on individual blocks in a secured manner using cryptographic hash functions.
Quality Assurance	Blockchain enables maintaining the quality of a dairy product throughout the supply chain. The use of blockchain assists in enforcing regulatory standards concerning the production, distribution, and storage of dairy products. Quality parameters of a particular dairy product can be tracked and maintained using blockchain implementation in its supply chain. Quality certificates are also generated using the blockchain network.
Decentralized	Blockchain-enabled supply chains are decentralized in nature thereby preventing any possibility of a single point of failure. Moreover, the decentralized nature prevents chances of data manipulation and spreading misinformation to other stakeholders.
Automation	Blockchain integration supports the highest levels of automation in the functioning of the supply chain. Updation of product information, payment settlements, removing a product, adding a stakeholder, and all functionalities are automated using the blockchain.
Removing Intermediaries	Unlike traditional supply chains, blockchain-enabled supply chains are devoid of intermediaries. Transactions are performed only between legitimate stakeholders ensuring the safety of the dairy products. Unauthorized stakeholders are not permitted to perform transactions or even enter their products into the supply chain.

**Table 5 foods-11-02716-t005:** Smart Contract Functions Table.

Function Name	Function Description
addproduct()	Add a new product
updateProduct_Info()	Update product information
getProduct_Info()	Retrieve all information with respect to a particular product. Trace history, billing information, adulterants, contaminants, nutrient values, expiry date, etc.
addProduct_Contamination_Info()	Add details of contamination tests over a particular product
getProduct_Contamination_Info()	Retrieve information regarding the nature and type of contamination present in a particular product
addProduct_Nutrition_Info()	Retrieve information with regard to nutritional values of a particular product
updateProduct_Quantity()	Update the number of products being produced by the company
getProduct_Qulaity()	Retrieve information with regard to quality parameters and nutritional values of a particular product
addDistributor()	Add a new distributor to the system
addRetailer()	Add a new retailer to the system
addCustomer()	Add a new customer to the system
addFarm()	Add a new dairy farm to the system
addShipmentInfo()	Add information regarding traceability of a product.
addProduct_Adulteration_Info()	Add details of adulteration tests performed over a particular product
getProduct_Adulteration_Info()	Retrieve information regarding the nature and types of adulterants present in a particular product
getProduct_Status()	Retrieve product information with regard to its presence in the supply chain
addProduct_Sales_Info()	Add sales related information specific to an individual product, product type, distributor, and retailer
getProduct_Sales_Info()	Returns sales related information specific to an individual product, product type, distributor, and retailer
addBilling_Info()	Add information relating to payments and other financial settlements with respect to a particular set of products between the dairy company and other stakeholders
getBilling_Info()	Returns information relating to payments and other financial settlements with respect to a particular set of products between the dairy company and other stakeholders
addProduct_Reuse_Info()	Add details of a particular product which can be reused within the supply chain
getProduct_Reuse_Info()	Retrieve details of a particular product which can be reused within the supply chain
getDistributor()	Returns details of a particular distributor
getRetailer()	Returns details of a particular retailer
getCustomer()	Returns details of a particular customer
getFarm()	Returns details of a particular dairy farm
addProcessingCentre()	Add a new processing centre to the system
getProcessingCentre()	Returns details of a particular processing centre
addColdStorage()	Add a new cold storage to the system
getColdStorage()	Returns details of a particular cold storage
getProduct_Stakeholder_Info()	Returns details of different product types and quantities with respect to a particular stakeholder

**Table 6 foods-11-02716-t006:** Blockchain Adoption Challenges for Dairy Supply Chain Management.

Adoption Challenge	Description
Skill Gap	Blockchain is still an emerging technology, therefore, leading to a shortage of skilled individuals who can work and create blockchain-based solutions. Existing employees in most IT companies are unaware of blockchain development tools and thus require proper training for becoming subject matter experts.
Interoperability	Organizations adopting blockchain tend to create proprietary solutions which are incompatible with other contemporary solutions. The interconnection of multiple blockchains for a single problem statement is still an open research area.
Inter-organization Trust Issues	The dairy supply chain comprises various stakeholders. Blockchain implementation will require the consensus of all stakeholders for the validation of transactions. The stakeholders need to trust one another for ensuring the proper functioning of the blockchain. In most cases, multiple stakeholders display a lack of trust concerning communication and coordination activities. Henceforth, building consensus thus turns out to become a huge challenge.
Lack of Standardization	Standards are essential in establishing trust and confidence in technology within the community. Blockchain technology lacks standards that facilitate inter-domain, inter-organisations, and inter-country transactions.
Scalability Issues	One of the biggest impediments to blockchain technology is its inability to scale. The number of transactions conducted per second is significantly lower for Ethereum when compared to Viva or PayPal. The lack of scalability for blockchain solutions is a significant roadblock to its ambitions of real-world implementations.
Financial Resources	Blockchain solutions require significant financial resources for infrastructure building and operations. The high computational cost for mining and validation of blocks is one of the major concerns for blockchain adoption.
Regulatory Compliance	Significant uncertainty prevails among customers when dealing with blockchain-based solutions. Governments of numerous developing countries are highly sceptical concerning the implementation of blockchain solutions. Proper legislations are still underway that ensures the protection of citizen rights in cases of using blockchain-based solutions. The use of cryptocurrencies is still considered illegal in most parts of the world.
Unregistered Stakeholders	Most developing countries such as India have multiple local stakeholders which are difficult to track and register on the system. Onboarding all such local stakeholders is a huge challenge for the implementation of a blockchain-based supply chain management platform.

## Data Availability

Data is contained within the article.

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
