# Peer review of "Blockchain-Enabled Supply Chain platform for Indian Dairy Industry: Safety and Traceability"

_foods, 2022, doi:10.3390/foods11172716_

Round 1

Reviewer 1 Report

Blockchain is an interesting and hot topic for safety management in the food chain. However, the framework of the paper is chaotic and more engineering-based, less academic contribution analysis. I suggest the authors can improve the quality about the current work. Major revisions.

1.      Please rewrite the Abstract. In general, the abstract is a condensation of the information (facts) in the paper and should present briefly and clearly the methods of the research and the principal results.

2.      Structural issue. The framework of the paper is chaotic. Literature review section should be behind the introduction. And Research Methodology is too simple. You should illustrate that how to conducts literature review, architecture design framework, development and applications on dairy industry, etc. About how to improve the logic framework of the paper, I suggest authors can refer the literature (https://doi.org/10.1016/j.jclepro.2020.121031)

3.      In section 4: overview of the Indian dairy industry, you should sufficiently justify why blockchain works better than other method for dairy industry traceability. AND this part should be background and combine with introduction or methodology section by less description.

4.      I suggest that the authors should give more emphasize on the challenges of implementing blockchain technology. Currently, blockchain is still not at a mature stage and thus have many challenges in the implementation, so I expect the authors may talk more about this point.

5.      Do not use an abbreviation without explaining it beforehand. The first time the word/phrase is introduced, put the abbreviation in brackets after the word/phrase. Some examples:

- in line , it 98appears for the first time "IoT" but the meaning of the abbreviation (Internet of Things) only is mentioned latter;

- the same for "RFID", "QR", and so on...

7. In section 7 smart contracts section, it is too engineering-based, not use to provide all of codes. I think the authors should presented the flow chart of smart contracts design, I suggest the authors can reduce the code description and add design and implement smart contracts logical framework. Please refer the literature https://doi.org/10.1016/j.compag.2021.10 6642. And make the paper more academic.

8. The authors presented experimentation setup, but lack the final test results. I suggest the authors can present real application and test result of blockcain traceability platform.

9.     The all of description needs to be improved, and the English of this manuscript also must be improved.

10. Some references are incomplete,please check references.

Author Response

At the beginning I would like to thank Reviewers for their time and valuable contributions. In attachment, you will find respond cover letter and the revised paper. All changes in the manuscript were highlighted using the “Track Changes” function.

Reviewer 2 Report

The paper is exciting. However, I have some comments to improve it. The comments are:

1.       In the introduction, it is necessary to discuss the unique characteristics of the dairy industry supply chain in India, which will determine the novelty of the substance of this article and should be supported by a solid literature review.

2.       In the overview of the dairy industry in India, it is better to discuss the uniqueness of the dairy supply chain in India, which determines the configuration or architecture of the blockchain to be developed, both from the aspect of supply chain structure, process, and management.

3.       In section 6.1. phases layer, it is not explained how the blockchain architecture developed accommodates the supply chain structure of the dairy industry in India involving small farmers and cooperatives.

4.       Does the blockchain architecture develop accommodate reverse logistics to ensure that there is no error in returning milk cans belonging to farmers from the collection point to each farmer?

5.       Why does the blockchain architecture develop not accommodate sources of contamination risk along the dairy industry supply chain? Thus, the blockchain platform can mitigate and reduce the possible contamination.

6.       Why does the developed smart contract not accommodate contracts between cooperatives as representatives of farmers and companies? It is necessary to reduce transaction costs.

7.       What justification for using multiple case studies in developing this blockchain technology? Is each case study unique?

8.       In the discussion section, it is necessary to conduct an in-depth analysis and discussion of the research findings, which are compared with the relevant research findings. Thus, this section will strongly contribute to theoretical and technological development.

9.       In conclusion, it is necessary to convey the limitations of this research and the development of future research that can be done.

Author Response

(The authors gave the same response as above.)

Reviewer 3 Report

The manuscript is more like a report for industrial application and the total page and number of words are exceeding a research paper. Furthermore, the IMRAD structure did not observe the work. The software which was used for the blockchain modeling must also be provided.

However, it is an interesting work on an industrial scale, I could not consider it a research paper.

Author Response

Dear Reviewer,

To begin with, we appreciate your work as a reviewer on our paper. It helped us to look at the article from a different angle, hence to improve it.  The revised paper is in attachment, changes in the manuscript are highlighted using the “Track Changes” function.

Concerning to exceeded volume of the paper, we are faced with two problems. First, reducing the manuscript volume may impair the presentation of the material and some information may be lost to the reader. Second, the rest reviewers have not seen a problem in the volume and even asked to add more information on some aspects. So, please, the question of the manuscript volume is ambivalent. However, to try to ameliorate the situation, we have decided to move some technical information to Appendix A, namely all smart contracts created and deployed for ensuring the functioning of the proposed traceability platform. Please, refer to lines 546-548 and lines 842-1170.

Concerning to IMRAD structure, we believe that our manuscript is created as per the” Introduction, Methods, Results, and Discussion” format because there are all essential sections in the paper. The Section 1 describing the Introduction, Section 2 as mentioning the Research Methodology, Section 3 presenting the Literature Review, Section 10 depicts the Results and Experimental Setup and finally Section 11 illustrating the Discussion and Future works.

Concerning to used software, the Ethereum blockchain was applied for building the traceability platform. Other software that was used during the development phase are Ganache, and VS Code. Ethereum blackchin was used for creating the blockchain network whereas smart contracts were created and tested using VS Code and Ganache respectively.  We have mentioned this information in Section 10. Please, refer to lines 767-787.

Concerning to the last comment, we disagree with you about your statement that our article is not research one. As we mentioned above, we believe that the paper is designed in accordance with IMRAD structure. The information on smart contracts were also moved to Appendix A to avoid the technical appearance of paper. The novelty of the paper is presented. It ensures the safety and traceability of dairy products through the supply chain thereby preventing its customers from consuming counterfeit products. The work illustrates a combination of technologies like blockchain, IoT, Cloud computing and machine learning working together in identifying and removing contaminated and adulterated dairy products from a supply chain. The work extends its novelty across four dimensions namely social, economic, operational and sustainability. The proposed work was successful in addressing all research questions enlisted at the start of the paper. A validation to its novelty the work has been compared with some of its prominent contemporaries. Please, refer to lines 194, 768 – 791.

Again, Thank you for your work. We hope that we could satisfy your concerns by our answers.

Sincerely, Authors

Round 2

Reviewer 1 Report

the structure of the paper is chaos and confused, no clear conceptual framework to describe the system development and evaluation.

Author Response

Dear Reviewer,

Thanks for the thorough review. We appreciate your comments, your work has helped us greatly improve the article better. In order to improve the structure of the paper, the following steps have been done:

  • We have reduced the number of pages for the paper by placing tables of Section 9 as part of Appendix A along with other amendments
  • We have enhanced the clarity of the paper by adding a graphical abstract and a flowchart describing the structure of the paper.
  • We have added the term Indian to the title

Reviewer 2 Report

The manuscript has been sufficiently improved.

Author Response

Dear Reviewer,

Thanks for the thorough review. We appreciate your comments, your work has helped us greatly improve the article better

Reviewer 3 Report

No further comments

Author Response

Dear Reviewer,

Thanks for the thorough review. We appreciate your comments, your work has helped us improve the article better